# Therapeutic Strategies for Overcoming Immunotherapy Resistance Mediated by Immunosuppressive Factors of the Glioblastoma Microenvironment

**DOI:** 10.3390/cancers12071960

**Published:** 2020-07-19

**Authors:** Tsubasa Miyazaki, Eiichi Ishikawa, Narushi Sugii, Masahide Matsuda

**Affiliations:** 1Department of Neurosurgery, Faculty of Medicine, University of Tsukuba, 1-1-1 Tennodai, Tsukuba, Ibaraki 305-8575, Japan; t-miyazaki@cell-medicine.com (T.M.); narushi-sugii@md.tsukuba.ac.jp (N.S.); m-matsuda@md.tsukuba.ac.jp (M.M.); 2Cell-Medicine, Inc., Sengen 2-1-6, Tsukuba Science City, Ibaraki 305-0047, Japan

**Keywords:** glioma, immune-checkpoint molecules, immunosuppressive tumor microenvironment, M2-type macrophages, tumor vaccine

## Abstract

Various mechanisms of treatment resistance have been reported for glioblastoma (GBM) and other tumors. Resistance to immunotherapy in GBM patients may be caused by acquisition of immunosuppressive ability by tumor cells and an altered tumor microenvironment. Although novel strategies using an immune-checkpoint inhibitor (ICI), such as anti-programmed cell death-1 antibody, have been clinically proven to be effective in many types of malignant tumors, such strategies may be insufficient to prevent regrowth in recurrent GBM. The main cause of GBM recurrence may be the existence of an immunosuppressive tumor microenvironment involving immunosuppressive cytokines, extracellular vesicles, chemokines produced by glioma and glioma-initiating cells, immunosuppressive cells, etc. Among these, recent research has paid attention to various immunosuppressive cells—including M2-type macrophages and myeloid-derived suppressor cells—that cause immunosuppression in GBM microenvironments. Here, we review the epidemiological features, tumor immune microenvironment, and associations between the expression of immune checkpoint molecules and the prognosis of GBM. We also reviewed various ongoing or future immunotherapies for GBM. Various strategies, such as a combination of ICI therapies, might overcome these immunosuppressive mechanisms in the GBM microenvironment.

## 1. Introduction—Glioblastoma and Its Epidemiological Features

Glioblastoma (GBM) is the most common and lethal malignant brain tumor, classified as grade IV by the World Health Organization (WHO), and reportedly diagnosed in 12.0% of all brain tumor patients [1,2,3]. GBM occurs frequently in people aged 50 to 60 years and is more common in men. Even with the current standard adjuvant therapy using temozolomide (TMZ) and radiation therapy (RT), the median relapse-free survival and median overall survival (OS) of GBM patients are 6.9 months and 14.6 months, respectively, and the 5-year survival rate is less than 10% [4]. Furthermore, in case of recurrence after standard therapy, the median survival time after recurrence is only 1 to 10.8 months, even if conditions such as the Karnofski performance status (KPS), tumor site, tumor volume, and expression of tumor markers are taken into account [5,6,7]. New treatment strategies for GBM patients are being successively developed. For instance, clinical trial of new drugs such as carmustine-loaded polymers (Gliadel wafers) without TMZ [8] and an anti-vascular endothelial growth factor (VEGF) antibody (bevacizumab) added to standard therapies [9] did not significantly improve the OS of initially diagnosed GBM patients.

Many researchers are focusing on various types of tumor proliferation markers, gene mutations/methylations, and histone mutations of GBM, such as Ki-67; TP53 mutations; epidermal growth factor receptor (EGFR) amplification and its mutant EGFRvIII, isocitrate dehydrogenase (IDH) 1/2 mutation; telomerase reverse transcriptase (TERT) promoter mutation; histone H3 mutations; epigenetic modifications of O6-methylguanine-DNA methyltransferase (MGMT) promotor methylation; and 1p19q co-deletion as prognostic markers of malignant glioma [10,11].

In GBM, IDH1/2 mutations and MGMT methylation are already considered prognostic and therapeutic markers but further therapeutic efficacy indicators are desired [10,11,12]. Anaplastic astrocytomas (WHO grade III) typically have IDH1 mutations and MGMT promotor methylation is an important prognostic marker [13]. Anaplastic oligodendrogliomas (WHO grade III), in contrast, typically have 1p19q co-deletion in addition to IDH1 mutation and have a better prognosis than GBM, even when they pathologically show necrotic features [14]. Regarding the changes in molecular expression before and after recurrence of GBM, the expression of p53 and EGFRvIII are decreased in recurrent tumor cells [15,16]. 

## 2. Glioblastoma Immune Microenvironment

Immune status biomarkers, such as the pro-/anti-inflammatory phenotype, infiltration rate, and activation status of tumor-infiltrated lymphocytes (TILs) in the tumor microenvironment and expression of immunosuppressive factors by tumor cells, are considered to be prognostic for various cancers (e.g., breast, colorectal) [17,18]. 

Immunosuppression is caused by various mechanisms in the tumor microenvironment. Representative immunosuppressive mechanisms include: expression of inhibitory immune checkpoint molecules (ICMs); infiltration of immunosuppressive immune cells, including M2-type macrophages (M2Mφs), forkhead box protein P3 (FOXP3)-positive regulatory T cells (Tregs), myeloid-derived suppressor cells (MDSC), and regulatory B cells (Bregs); secretion of immunosuppressive cytokines; and reduced expression of MHC molecules [19]. In the glioma microenvironment, FASL (CD95L) [20], programmed cell death ligand 1/2 (PD-L1/2) [21,22,23,24,25], galectin-1 [26], galectin-9 [27], HVEM [28], B7-H4 (B7x) [29], and CD70/gangliosides [30] expressed on glioma cells act as inhibitory ICMs. Furthermore, AHR and CD39 expressed on TAMs [31], podoplanin (PDPN) expressed on MDSCs [32], and glycoprotein A repetitions predominantly (GARP) expressed on Tregs [33] also cause immunosuppression. As for humoral factors, VEGF [34], macrophage migration inhibitory factor (MIF) [35], IL-6 [29], IL-4/13 [36,37], IL-10, TGFβ [38], POSTN [39], colony-stimulating factor-1 (CSF-1) [40], CCL2 [41], CXCL12 [42], and COX-2/PGE2 [43] directly or indirectly regulate the immunosuppressive tumor microenvironment. Stat3, an intracellular transducer of immune-related signaling, acts to promote proliferation, angiogenesis, metastasis, and immune escape in tumor cells [44]. In addition, extracellular vesicles containing PD-L1 [45] and miRNA [46] secreted by glioma cells control the immunosuppressive ability of infiltrating immune cells. Alterations to the local nutritional status [47] by indoleamine 2,3 dioxygenase 1 (IDO) expression [48] or Tregs [49], CD39/CD73/A2AR [50], protein arginine methyltransferase 5 (PRMT5) [51], and HIF1α expression associated with hypoxia during tumor progression [52] also contribute to an immunosuppressive microenvironment surrounding tumor cells. IDO metabolizes tryptophan to kynurenine and induces/activates Tregs in the tumor microenvironment [48]. Furthermore, paracrine crosstalk between tumor cells and immunosuppressive cells, as well as between immunosuppressive cells themselves, augments immunosuppression in the tumor microenvironment [29,35,36,37,38,39,40,41,42,43,45,48,49,50,53]. Thus, GBM cells and their microenvironment facilitate immunosuppression through a variety of strategies and development of treatments with diversified/comprehensive immunological perspectives is desirable (Figure 1 and Figure 2).

## 3. Association between Expression of ICMs and Prognosis in GBM

The immune system is a biological defense mechanism that has evolved to prevent/eliminate the invasion of external enemies (such as bacteria and viruses) but it also eliminates tumor cells. The effector function of immune cells is controlled by ICM expression. When CD8 T cells attack target cells, the first signal of the MHC molecule carrying the target antigen followed by co-stimulative ICMs—such as CD28, ICOS, or CD134 (OX40)—is required [54]. The expression of NR4A1, which inhibits T cell effector function, is also an obstacle to immune activation [55] MHC-deficient cells are targeted and attacked by NK cells. In the absence of these co-stimulative ICMs, however, T cells lapse into an anergic, immune-deficient state. Activated CD8 T cells attack target cells one after another while controlling their own activity by the expression of co-inhibitory ICMs such as cytotoxic T-lymphocyte antigen-4 (CTLA-4), PD-1, and T-cell immunoglobulin and mucin domain 3 (TIM-3) to prevent excessive activation and autoimmune effector activity [54]. On the other hand, tumor cells also utilize these inhibitory ICMs for immune escape. Therefore, novel therapies focused on these inhibitory ICMs and T-cell exhaustion have recently been developed [56]. Among them, inhibitory antibodies of the PD-1/PD-L1 pathway have shown therapeutic effects in various types of cancers, especially in high somatic mutation burden tumors such as melanoma and non-small cell lung cancer [57,58]. The relationship between PD-1/PD-L1 expression and prognosis has been reported for several types of cancers [59].

The prognostic effect of PD-1/PD-L1 expression in the tumor microenvironment of GBM has also been analyzed. Using immunohistochemical analysis, Nduom and coauthors reported that PD-L1 expression was a poor prognosis marker in 94 GBM patients [25]. Similarly, in the analysis of 17 GBM patient samples, Liu and coauthors found that over 10 PD-L1-expressing cell phenotypes in tumor tissue were associated with poor prognosis [23]. However, in the analysis of PD-L1 mRNA expression in 135 specimens (117 initial specimens and 18 local recurrence specimens), no such correlations were found [21]. We have also previously reported that PD-1/PD-L1 expression in primary tumors does not correlate with GBM prognosis [24]. As described above, the correlation between PD-1/PD-L1 expression and patient prognosis is still controversial in GBM [21,23,25]. In addition, infiltration of Tregs and IDO1 expression were reported as poor prognostic factors [60,61]. Thus, such molecules in the tumor microenvironment are potential therapeutic targets [62].

## 4. GBM Immunotherapy and Microenvironmental Changes after Recurrence

Immunotherapy for malignant tumors originated in the 1890s, when Dr William Coley discovered the relationship between erysipelas infection and tumor disappearance [63]. Cytokine therapy that administers interleukins and interferons, adoptive immunotherapy that extracts and activates lymphocytes and dendritic cells from the blood, and artificially synthesized cancer vaccine therapy that administers cancer antigens or cancer tissue-processed products added with immune adjuvants have been developed [64]. Our group has previously reported a clinical trial using autologous tumor-specific T lymphocytes [65] and autologous natural killer cells [66] for recurrent malignant glioma. Another report on GBM antitumor therapy compared autologous formalin-fixed tumor vaccine (AFTV), manufactured from autologous formalin-fixed tumor tissue [64], a vaccine with RT [67] and TMZ concomitant with RT standard therapy [68]. In the most recent phase IIa clinical trial, the median OS was 22.2 months and the 3-year survival rate was 38% [68,69]. The number of TILs was increased in recurrent tumor tissues after AFTV therapy compared with initial tumor tissue and the number of Ki-67-positive tumor cells tended to be decreased [70]. Furthermore, the number of PD-1-positive inactivated/exhausted lymphocytes was increased in recurrent GBM tissue, especially in patients treated with AFTV before recurrence (Figure 3) [24].

Clinical trials on the anti-PD-1 antibody ICI nivolumab failed to show an advantage over control bevacizumab adjuvant therapy for recurrent GBM patients, indicating that the therapeutic effects of ICI alone appear to be limited to recurrent GBM [71]. On the other hand, ICIs may be effective if used as neoadjuvant therapy for resectable, recurrent GBM [72]. M2Mφ infiltration in the tumor microenvironment and the phosphatase and tensin homolog (PTEN) mutation status may affect the therapeutic effect of ICIs [73]. The efficacy of ICI treatment may thus be improved by combination with other treatments or by patient selection according to tumor mutation genotype. Improving the treatment efficacy of immunotherapy and establishing immune memory will lead to increased long-term survival of patients with GBM. Factors considered as combinational therapeutic targets are described below (Figure 3 and Figure 4).

## 5. Regulatory T Cells (Tregs) as a Therapeutic Target of GBM

Tregs that mainly differentiate from the naïve T-cell fraction to suppress excessive activation of T cells are described as CD25^+^ FOXP3^+^ CD4 T cells and function to suppress antitumor immunity by effector T cells in the tumor microenvironment [74]. Tregs are classified by molecular marker expression level and by examination of their functions; thus, it has been reported that the fraction of FoxP3^high^ CD45RA- CD25^high^ cells has strong immunosuppressive abilities, including expression of co-inhibitory ICMs, consumption of autoexpanding IL-2, production of inhibitory cytokines, consumption of nutrients, and killing activity against effector T cells [75]. For many types of malignant tumors, the proportion of Tregs in the tumor and peripheral blood and the ratio of Tregs to CD8 cells correlate with the prognosis [76,77,78,79,80,81,82,83]. In GBM, there are opposite reports regarding the relationship between Treg proportion and the CD8/Treg ratio with prognosis [84,85,86,87]. On the other hand, Treg accumulation in the tumor margins [88] and inhibitory ability in murine models have been reported [89], indicating that Tregs can also be therapeutic targets in GBM. Combination therapy targeting angiogenesis using VEGF and angiopoietin-2 inhibitors combined with anti-PD-1 antibody inhibited Treg and MDSC infiltration and increased proliferation and anti-tumor activity of glioma-infiltrating CD8 T cells [90]. Anti-CD25 antibody, anti-CCR4 antibody, anti-CXCR4 antibody, anti-CTLA-4 antibody, IL-10, and TGFβ inhibitors have been clinically developed as Treg depletion therapy (Figure 4) [91,92,93,94]. Combination therapy using Treg depletion with anti-CD25 or anti-CXCR4 antibodies and ICIs was shown to be effective in a murine glioma model [94,95]. Regarding Treg-depleting antibodies, methods of suppressing the onset of autoimmune diseases by adjusting the administration method and period are also under investigation [96,97]. However, these antibody therapies are systemically effective and may cause autoimmune adverse effects [75,96,97]. In addition, there are several types of Tregs: a fraction called FOXP3^low^ ‘fragile or non’ Tregs with low immunosuppressive function and a truly functional fraction called FOXP3^high^ ‘effector’ Tregs. Development of a therapy targeting these effector Tregs is required [96,97,98].

## 6. Myeloid-Derived Suppressor Cells (MDSCs), M2 Macrophages (M2Mφs), and Regulatory B Cells (Bregs) as Therapeutic Targets of GBM

MDSCs are detected as a fraction of CD11b^+^CD33^+^HLA-DR^−^ cells in humans and are classified into M-MDSC CD11b^+^CD33^+^HLA-DR^−^/CD14^+^CD15^−^ cells and PMN (polymorphonuclear)-MDSC CD11b^+^CD33^+^HLA-DR^−^/CD14^-^CD15^+^ cells [99]. The tumor infiltration and blood level of MDSCs correlate with the prognosis of many types of cancers [99]. MDSCs accumulate in the tumor microenvironment and suppress anti-tumor immunity by signals—such as cytokines, chemokines, and extracellular vesicles—secreted from tumor cells [100,101]. These bone marrow-derived cells cause immunosuppression of the tumor microenvironment not only by humoral factors such as IL-10 and TGFβ, but also by ICMs, extracellular vesicles, and nutrient consumption [101]. In GBM, several reports revealed increased MDSC levels in the blood or tumor tissue as a poor prognostic factor [84,102,103,104]. CCL2 produced by murine GBM cells recruits CCR4^+^Tregs and CCR2^+^Ly-6C^+^ mMDSCs [105] while MIF produced by tumor stem cells activates MDSCs, resulting in an immunosuppressive tumor microenvironment [105]. In addition, CD49d mRNA levels in tumor tissue, suggesting CD49d^+^ MDSC/TAM/Treg levels, strongly correlate with GBM prognosis [103]. On the basis of such reports, combination immunotherapy for MDSC inhibition has been developed. Representative examples are described below. Combination therapy using a CCR2 antagonist (CCX872) and anti-PD-1 antibody to inhibit MDSC recruitment into the tumor microenvironment for KR158 (HGG-like cells) and 005GSC (stem-like cells) murine models increases IFNγ^+^TILs and prolongs overall survival [106]. Combination therapy using anti-PD-1 antibody and an inhibitor of CXCR4, a receptor for CXCL12/SDF-1 that contributes to the maintenance of tumor stem cells, also increases local infiltration of CD4/8 T cells by suppressing Treg and MDSC tumor invasion and prolongs the survival of tumor-bearing mice [94]. CD200 is required to maintain bone marrow cell homeostasis; however, it may cause exacerbation due to increased MDSC infiltration during tumor growth. Therefore, combination of a CD200 synthetic peptide that leads to production of anti-CD200 antibodies in the organism and inhibits MDSCs with a tumor vaccine has been developed [107]. Since IL-6 produced by glioma cells promotes PD-L1 expression of MDSCs, the immunosuppressive function of MDSC is suppressed by IL-6 KO or an anti-IL-6 antibody and the antitumor effect on glioma cells is enhanced by combination therapy using anti-PD-1 antibodies [108]. αPD-L1-LNP, an anti-PD-L1 antibody-conjugated lipid nanoparticle (LNP) containing a CDK inhibitor (dinaciclib), demonstrated a therapeutic effect in murine glioma models. It eliminated tumor-associated bone marrow cells (TAMCs) localized to the tumor microenvironment after RT by inducing apoptosis [109]. Owing to the strong immunosuppressive potential of MDSCs and their prognostic impact on GBM, the development of a combination therapy to inhibit MDSC infiltration is desirable.

Among immunosuppressive cells, M2Mφs have a wide variety of functions that promote tumor growth and therefore cause strong immunosuppression when recruited into the tumor microenvironment at the early stage of tumor growth [110]. M2Mφs enhance PD-L1 and IDO expression by antibody-dependent cellular phagocytosis [111]. It is expected that the effects of ICIs can be enhanced by combination with an M2Mφ inhibitor. In GBM, B7-H4 expression on tumor cells and tumor-associated macrophages (TAMs) results in the maintenance of glioma progenitor cells and the formation of an immunosuppressive tumor microenvironment [29] while miRNA-21 contained in exosomes produced by TAMs increases the production of PDCD4, SOX2, STAT3, IL-6, and TGF-β1 in GBM cells that engenders TMZ resistance [112]. Conversely, exosomes derived from glioma stem cells (GSCs) promote M2 polarization and PD-L1 expression in Mφs [113]. Hence, tumor cells and Mφs mutually regulate the survival environment. Additionally, large amounts of Mφs infiltrate GBM tissues and are derived from monocytes rather than microglia [114,115]. IDO expressed by TAMs and tumor cells metabolizes tryptophan to kynurenine, and kynurenine inhibits T-cell immunity while stimulating Mφ aryl hydrocarbon receptor (AHR) expression. This AHR signal increases CCR2 and CD39, an ATP/ADP-degrading ectonucleotidase, expression on TAMs, resulting in increased CCL2-induced TAM recruitment into the tumor microenvironment and subsequent rise in the environmental adenosine concentration by the ectonucleotidase activity of CD39/CD73; all of which contributes to tumor progression by suppressing T-cell immunity [31]. CD73, an AMP-degrading ectonucleotidase that functions in cooperation with CD39, is highly expressed by TAMs infiltrating GBM and is expected to be a target for combination immunotherapy [116]. In addition to treatments aimed at inhibiting the function of Mφs, treatments aimed at inhibiting Mφs accumulation or M1 conversion have also been developed. However, a colony-stimulating factor-1 receptor (CSF-1R) inhibitor aimed at obstructing accumulation of Mφs that did not alter the infiltration number of TAMs suppressed M2 function and improved prognosis in a murine glioma model [117]. In addition, combination therapy using a dendritic cell (DC) vaccine and an anti-PD-1 antibody with the CSF-1R inhibitor prolonged OS in a murine model via promotion of infiltration and activation of TIL [118] IPI-549, an inhibitor of phosphatidylinositol 3-kinase γ (PI3Kγ), an intracellular signal of M2Mφs, suppresses M2Mφs by selectively inhibiting PI3Kγ (IC_50_:16 nM) and inducing the PI3Kδ-dominant M1 phenotype [119]. Tumor growth inhibitory effects of IPI-549 have been confirmed in combination with anti-PD-1 antibodies in lung, breast, and head/neck cancer models [120]. We have also demonstrated the antitumor effect of combination therapy using anti-PD-L1 antibody and IPI-549 in a TMZ-resistant glioma-initiating murine model [121]. Furthermore, using human GBM tissue, we revealed an M2Mφ infiltration increase during recurrence after intervention with immunotherapy as compared with the primary tumor [121]. Triple combination therapy using oncolytic herpes simplex viruses (oHSV, G47D) expressing murine IL-12 (G47D-mIL12) with anti-CTLA-4 and anti-PD-1 antibodies also induces Mφ infiltration and M1-like phenotype polarization, contributing to the eradication rate in a murine model using GSCs via increasing the CD8^+^/CD4^+^FOXP3^+^ ratio [122].

By receiving extracellular vesicles produced by MDSC and M2Mφs, B cells express inhibitory ICMs such as PD-L1 or CD155, thereby forming an immunosuppressive tumor microenvironment as Bregs [53]. Tumor growth was significantly suppressed by using CD20 antibody or B-cell knockout in a glioma murine model in vivo [53]. Since extracellular vesicles derived from MDSC or M2Mφs control the immunosuppressive ability of Bregs, a therapeutic strategy for targeting MDSC, M2Mφs, or extracellular vesicles may be effective.

## 7. Ongoing Clinical Trials

A search of the NIH website (https://clinicaltrials.gov/) conducted on January 27, 2020 for ongoing immunotherapy clinical trials for GBM revealed 61 clinical trials. Excluding observational and non-interventional studies, 58 were intervention trials related to immunotherapy for GBM (Table 1). Registration status is 14 for “Active, not recruiting” and 2 for “Active, not recruiting / Has results”, 2 for “Enrolling by invitation” and 39 for “Recruiting”. Twenty-two of these trials were ICI-related and 11 were in combination with other immunotherapies such as adoptive immunotherapy including dendritic cell therapy, oncolytic virus therapy, IDO inhibitors, and vaccine therapy. Although there is some overlap with the abovementioned, there were 21 adoptive immunotherapies including dendritic cell therapy, TIL therapy, and CAR-T therapy; 11 oncolytic virus therapies; 4 IDO inhibitors; and 5 vaccine therapies. 

Numerous phase I-II clinical trials of state-of-the-art treatments, such as a DC vaccine, ICI combination, chimeric antigen receptor (CAR)-T cell therapy, a cytomegalovirus (CMV) pp65 vaccine, and recombinant human IL-7-hybrid Fc NT-I7 (IL-7-hyFc, GX-I7), are ongoing. Phase III clinical trials included nivolumab and phase II/III clinical trials included DC therapy. Among these, immunosuppressive cells, including Tregs, MDSC, and M2Mφs in the tumor microenvironment, are often the standard therapeutic target; however, a search of the UMIN website for clinical trials (https://upload.umin.ac.jp/) revealed that, in Japan, no active clinical trials related to immunotherapy for GBM are currently being conducted.

## 8. Future Perspectives and Conclusions

The most crucial point in the treatment of GBM is recurrence prevention after surgery and the development of immunotherapies to expand treatment options is welcomed. However, single immunotherapy is often insufficient in GBM and leads to formation of a ‘cold tumor’, since the immune microenvironment in GBM varies so greatly compared to other types of cancers. Analysis of the GBM microenvironment shows that patients with poor prognosis often have infiltration of immunosuppressive cells such as Tregs, MDSC, and M2Mφs (Figure 3). A neoantigen vaccine using comprehensive gene analysis [123], CAR-T/NK cell therapy [124,125,126], and oncolytic virus therapy [122,127] are also considered promising. In addition to combinations of chemotherapies, radiotherapies, and ICIs already available for other malignant tumors [128], these ongoing or future therapies targeting the above-mentioned cells or immunosuppressive function will be forthcoming. It is also important to establish minimally invasive measurement methods to assay changes in immune status, such as liquid biopsy using the blood or cerebrospinal fluid of GBM patients [129].

In conclusion, we have here reviewed epidemiological features, tumor immune microenvironment, and associations between ICM expression and GBM prognosis. We have also reviewed the various ongoing and future immunotherapies for GBM. Various strategies, such as combinations of ICI therapies, will overcome these immunosuppressive mechanisms in the immune microenvironment of GBM.

## Figures and Tables

**Figure 1 cancers-12-01960-f001:**
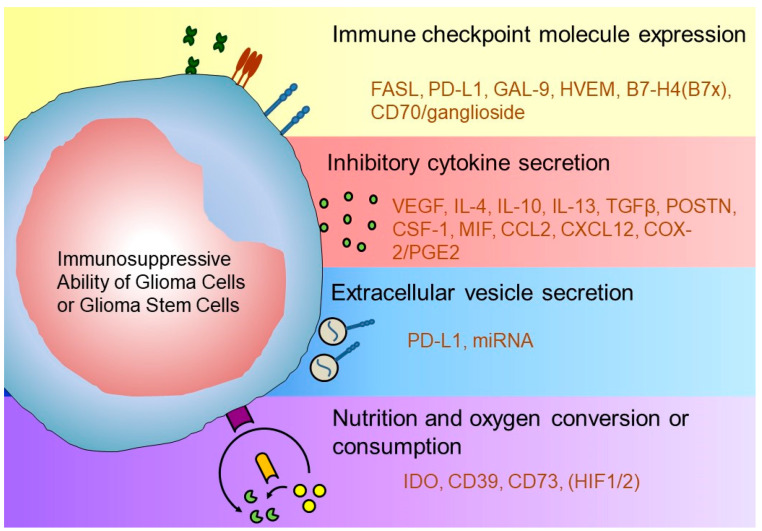
Glioblastoma cell immunosuppressive ability and cell-to-cell interactions in the glioma microenvironment. The immunosuppressive tumor microenvironment is created by altering the immune checkpoint molecule expression, immunosuppressive cytokine secretion, extracellular vesicle secretion, and oxygen nutrition status.

**Figure 2 cancers-12-01960-f002:**
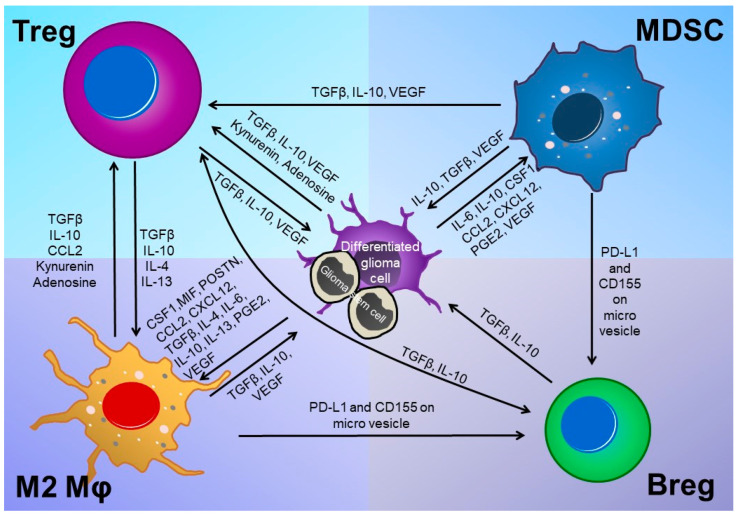
Complex crosstalk among tumor cells and various immune cells in glioma microenvironment. In the glioma microenvironment, the immunosuppressive environment is reinforced by a complex crosstalk between tumor cells and immunosuppressive cells and between different immunosuppressive cells.

**Figure 3 cancers-12-01960-f003:**
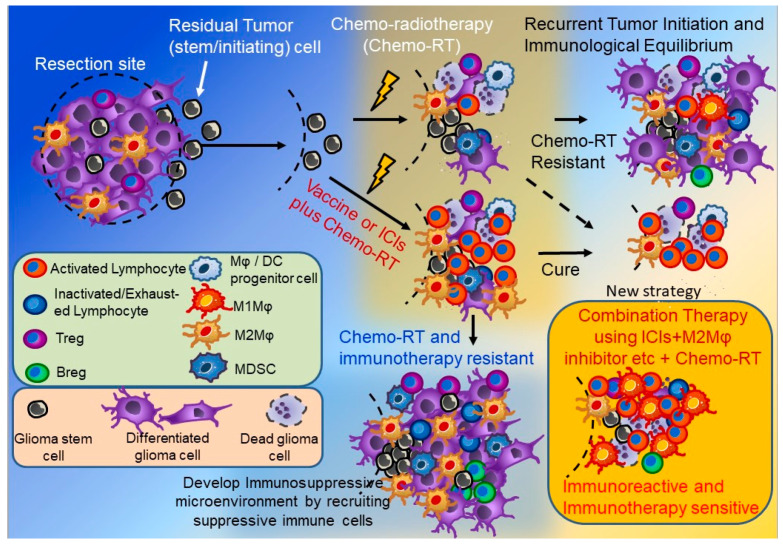
Therapeutic strategies for controlling immunosuppressive cells and ICI combination therapy for glioma. The immunosuppressive cell number increases after standard therapy +/− ICIs in the glioma microenvironment and the tumor becomes ‘cold’ (blue area). ICIs plus anti-immunosuppressive cell combination therapy causes the tumor to become ‘hot’ (orange area).

**Figure 4 cancers-12-01960-f004:**
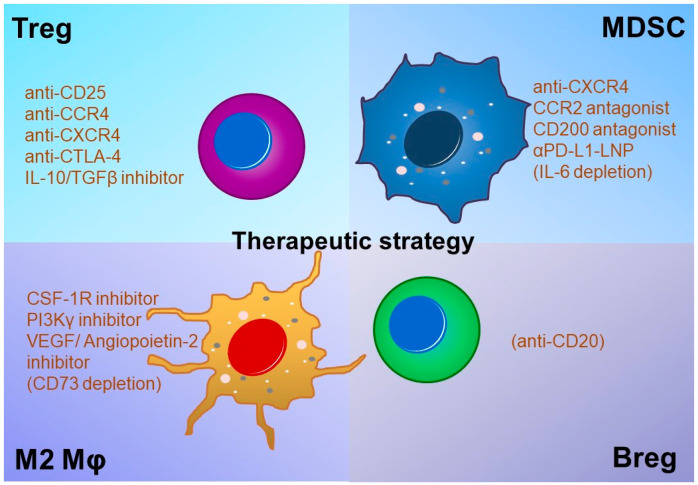
Treatment candidates for Tregs, MDSCs, M2Mφ, and Bregs in the glioma microenvironment.

**Table 1 cancers-12-01960-t001:** Ongoing clinical trials

Status/Phase	Study Title	Interventions	NCT Number
N/A/I	Nivolumab With DC Vaccines for Recurrent Brain Tumors	Nivolumab + dendritic cell vaccine	NCT02529072
1/II	Autologous Lymphoid Effector Cells Specific Against Tumour (ALECSAT) as Add on to Standard of Care in Patients with Glioblastoma	ALECSAT + radiotherapy + temozolomide	NCT02799238
1/I	Vaccine Therapy in Treating Patients with Newly Diagnosed Glioblastoma Multiforme	Tetanus toxoid + therapeutic autologous dendritic cells + therapeutic autologous lymphocytes	NCT00639639
1/II	Pembrolizumab in Treating Patients with Recurrent Glioblastoma	Pembrolizumab	NCT02337686
1/I/II	INO-5401 and INO-9012 Delivered by Electroporation (EP) in Combination with Cemiplimab (REGN2810) in Newly-Diagnosed Glioblastoma (GBM)	DNA-based cancer vaccine INO-5401(hTERT, WT1, and PSMA) + INO-9012 (IL-12) + cemiplimab + radiotherapy + temozolomide	NCT03491683
1/I	DNX-2440 Oncolytic Adenovirus for Recurrent Glioblastoma	DNX-2440 (Oncolytic Adenovirus)	NCT03714334
1/I	A Pilot Surgical Trial to Evaluate Early Immunologic Pharmacodynamic Parameters For The PD-1 Checkpoint Inhibitor, Pembrolizumab (MK-3475), In Patients With Surgically Accessible Recurrent/Progressive Glioblastoma	Pembrolizumab	NCT02852655
1/II	Avelumab in Patients with Newly Diagnosed Glioblastoma Multiforme	Avelumab	NCT03047473
1/II	Dendritic Cell Vaccine for Patients with Brain Tumors	Autologous tumor lysate-pulsed DC vaccination + adjuvant polyICLC	NCT01204684
1/II	Avelumab With Hypofractionated Radiation Therapy in Adults with Isocitrate Dehydrogenase (IDH) Mutant Glioblastoma	Avelumab + hypofractionated radiation therapy (HFRT)	NCT02968940
1/II	Convection-Enhanced Delivery (CED) of MDNA55 in Adults with Recurrent or Progressive Glioblastoma	MDNA55 (a fusion protein comprising a genetically engineered Interleukin-4 (IL-4) linked to a modified version of the *Pseudomonas aeruginosa* exotoxin A (PE))	NCT02858895
1/I	Phase I Study of a Dendritic Cell Vaccine for Patients with Either Newly Diagnosed or Recurrent Glioblastoma	Dendritic cell vaccine + radiotherapy + temozolomide ± bevacizumab	NCT02010606
1/II	Tremelimumab and Durvalumab in Combination or Alone in Treating Patients with Recurrent Malignant Glioma	Durvalumab + tremelimumab	NCT02794883
1/II	Combination Adenovirus + Pembrolizumab to Trigger Immune Virus Effects	DNX-2401 (Oncolytic adenovirus) + pembrolizumab	NCT02798406
1/I	A Phase I Study of AdV-tk + Prodrug Therapy in Combination with Radiation Therapy for Pediatric Brain Tumors	AdV-tk (an adenoviral vector (disabled virus) engineered to express the Herpes thymidine kinase gene) + valacyclovir + radiation	NCT00634231
2/II	Bevacizumab with or Without Trebananib in Treating Patients With Recurrent Brain Tumors	Bevacizumab + trebananib	NCT01609790
2/II	Phase 2 Study of Durvalumab (MEDI4736) in Patients with Glioblastoma	Durvalumab + radiotherapy + temozolomide + bevacizumab	NCT02336165
3/II/III	Proteome-Based Personalized Immunotherapy of Glioblastoma	Dendritic vaccine + allogeneic hematopoietic stem cells + cytotoxic lymphocytes	NCT01759810
3/I	Immunogene-modified T (IgT) Cells Against Glioblastoma Multiforme	Antigen-specific IgT cells	NCT03170141
4/I/II	Adjuvant Dendritic Cell-immunotherapy Plus Temozolomide in Glioblastoma Patients	Dendritic cell vaccine + temozolomide	NCT02649582
4/II/III	Dendritic Cell Immunotherapy Against Cancer Stem Cells in Glioblastoma Patients Receiving Standard Therapy	Dendritic cell immunization + adjuvant temozolomide	NCT03548571
4/II	Immunotherapy Targeted Against Cytomegalovirus in Patients with Newly-Diagnosed WHO Grade IV Unmethylated Glioma	Human CMV pp65-LAMP mRNA-pulsed autologous DCs containing GM CSF + temozolomide + tetanus–diphtheria toxoid (Td) 111-Indium-labeling of Cells for in vivo Trafficking Studies	NCT03927222
4/II	V-Boost Immunotherapy in Glioblastoma Multiforme Brain Cancer	V-Boost (an oral tablet which contains specially formulated hydrolyzed GBM antigens along with alloantigens)	NCT03916757
4/II	Study of DC Vaccination Against Glioblastoma	Dendritic cell vaccine + radiotherapy + temozolomide	NCT01567202
4/I	Pembrolizumab and Vorinostat Combined with Temozolomide for Newly Diagnosed Glioblastoma	Pembrolizumab + vorinostat + temozolomide + radiotherapy	NCT03426891
4/III	An Investigational Immuno-Therapy Study of Temozolomide Plus Radiation Therapy with Nivolumab or Placebo, for Newly Diagnosed Patients with Glioblastoma (GBM, a Malignant Brain Cancer)	Nivolumab + temozolomide + radiotherapy	NCT02667587
4/III	An Investigational Immuno-Therapy Study of Nivolumab Compared to Temozolomide, Each Given with Radiation Therapy, for Newly-diagnosed Patients With Glioblastoma (GBM, a Malignant Brain Cancer)	Nivolumab + temozolomide + radiotherapy	NCT02617589
4/I	Biomarker-Driven Therapy Using Immune Activators with Nivolumab in Patients with First Recurrence of Glioblastoma	Nivolumab + anti-GITR monoclonal antibody MK-4166 + IDO1 inhibitor INCB024360 + ipilimumab	NCT03707457
4/II	Radiation Therapy Plus Temozolomide and Pembrolizumab with and without HSPPC-96 in Newly Diagnosed Glioblastoma (GBM)	Pembrolizumab + HSPPC-96 (an autologous tumor-derived heat shock protein peptide-complex) + temozolomide	NCT03018288
4/I	Nivolumab, BMS-986205, and Radiation Therapy with or without Temozolomide in Treating Patients with Newly Diagnosed Glioblastoma	IDO1 Inhibitor BMS-986205 + nivolumab + radiation therapy + temozolomide	NCT04047706
4/I	Pembrolizumab and a Vaccine (ATL-DC) for the Treatment of Surgically Accessible Recurrent Glioblastoma	Dendritic cell tumor cell lysate vaccine + pembrolizumab + poly ICLC	NCT04201873
4/I	Genetically Modified T-cells in Treating Patients with Recurrent or Refractory Malignant Glioma	IL13Rα2-specific, hinge-optimized, 41BB-costimulatory CAR/truncated CD19-expressing Autologous T lymphocytes	NCT02208362
4/I	IL13Ralpha2-Targeted Chimeric Antigen Receptor (CAR) T Cells with or without Nivolumab and Ipilimumab in Treating Patients with Recurrent or Refractory Glioblastoma	IL13Ralpha2-specific hinge-optimized 4-1BB-co-stimulatory CAR/Truncated CD19-expressing autologous TN/MEM cells + ipilimumab + nivolumab	NCT04003649
4/I/II	Atezolizumab in Combination with Temozolomide and Radiation Therapy in Treating Patients with Newly Diagnosed Glioblastoma	Atezolizumab + radiation therapy + temozolomide	NCT03174197
4/II	Immunotherapy Using Tumor Infiltrating Lymphocytes for Patients with Metastatic Cancer	Young TIL + aldesleukin + cyclophosphamide	NCT01174121
4/I/II	NCT Neuro Master Match - N²M² (NOA-20)	APG101 (a soluble CD95-Fc fusion protein) or alectinib or idasanutlin or atezolizumab or vismodegib or palbociclib	NCT03158389
4/II	Efficiency of Vaccination with Lysate-loaded Dendritic Cells in Patients with Newly Diagnosed Glioblastoma	autologous, tumor lysate-loaded, mature dendritic cells (DC) + radiation therapy + temozolomide	NCT03395587
4/I	Memory-Enriched T Cells in Treating Patients with Recurrent or Refractory Grade III-IV Glioma	CD19CAR-CD28-CD3zeta-EGFRt-expressing Tcm-enriched T-lymphocytes + CD19CAR-CD28-CD3zeta-EGFRt-expressing Tn/mem-enriched T-lymphocytes	NCT03389230
4/I/II	A Phase I/IIa Study Evaluating Temferon in Patients with Glioblastoma and Unmethylated MGMT	Temferon	NCT03866109
4/I	Phase I EGFR BATs in Newly Diagnosed Glioblastoma	EGFR BATs + radiation therapy + temozolomide	NCT03344250
4/I	Adoptive Cell Therapy of Autologous TIL and PD1-TIL Cells for Patients with Glioblastoma Multiforme	Autologous TIL+ PD1-TIL	NCT03347097
4/II	Pediatric Trial of Indoximod With Chemotherapy and Radiation for Relapsed Brain Tumors or Newly Diagnosed DIPG	Indoximod + partial radiation or full-dose radiation	NCT04049669
4/I	Combination of Immunization and Radiotherapy for Malignant Gliomas (InSituVac1)	GM-CSF + Poly I:C or CAR-T or TCR-T + radiation	NCT03392545
4/I	Avelumab With Laser Interstitial Therapy for Recurrent Glioblastoma	Avelumab + MRI-guided LITT therapy	NCT03341806
4/I	Genetically Engineered HSV-1 Phase 1 Study for the Treatment of Recurrent Malignant Glioma	M032 (NSC 733972) (a second-generation oncolytic herpes simplex virus (oHSV))	NCT02062827
4/II	Non-Viral TCR Gene Therapy	fludarabine + cyclophosphamide + aldesleukin + sleeping beauty transposed PBL	NCT04102436
4/I	Safety and Immunogenicity of Personalized Genomic Vaccine and Tumor Treating Fields (TTFields) to Treat Glioblastoma	Poly-ICLC + tumor treating fields + peptides vaccine	NCT03223103
4/I	A Study to Evaluate the Safety, Tolerability and Immunogenicity of EGFR(V)-EDV-Dox in Subjects with Recurrent Glioblastoma Multiforme (GBM)	EGFR(V)-EDV-Dox (a bacterially derived minicell which packages a toxic payload, doxorubicin, into a 400 nm particle which targets specific cancer cells using bispecific antibodies (BsAb))	NCT02766699
4/II	Administration of Autologous T-Cells Genetically Engineered to Express T-Cell Receptors Reactive Against Mutated Neoantigens in People with Metastatic Cancer	Cyclophosphamide + fludarabine + aldesleukin + individual patient TCR-Transduced PBL + pembrolizumab	NCT03412877
4/I/II	Dose-Escalation Study to Evaluate the Safety and Tolerability of GX-I7 in Patients with Glioblastoma	GX-I7 (a protein drug recombining human IL-7 and hybrid Fc (hyFc))	NCT03619239
4/I/II	Study to Evaluate Safety, Tolerability, and Optimal Dose of Candidate GBM Vaccine VBI-1901 in Recurrent GBM Subjects	VBI-1901 (a polyvalent therapeutic vaccine against cytomegalovirus antigen gB and pp65) + GM-CSF	NCT03382977
4/I	Trial of C134 in Patients with Recurrent GBM	C134 (a cancer killing virus (HSV-1))	NCT03657576
4/I	GMCI, Nivolumab, and Radiation Therapy in Treating Patients with Newly Diagnosed High-Grade Gliomas	AdV-tk + valacyclovir + radiation + temozolomide + nivolumab	NCT03576612
4/I	Study of the IDO Pathway Inhibitor, Indoximod, and Temozolomide for Pediatric Patients with Progressive Primary Malignant Brain Tumors	Indoximod + temozolomide + conformal radiation + cyclophosphamide + cyclophosphamide	NCT02502708
4/I	A Study of the Treatment of Recurrent Malignant Glioma With rQNestin34.5v.2	rQNestin (an oncolytic viral vector made from the herpes simplex virus type 1 (HSV1)) + cyclophosphamide	NCT03152318
4/I	Phase 1b Study PVSRIPO on Recurrent Malignant Glioma in Children	polio/rhinovirus recombinant (PVSRIPO)	NCT03043391
4/I	HSV G207 in Children with Recurrent or Refractory Cerebellar Brain Tumors	G207 (an oncolytic herpes simplex virus-1 (HSV))	NCT03911388
4/I	HSV G207 Alone or With a Single Radiation Dose in Children with Progressive or Recurrent Supratentorial Brain Tumors	G207 (an oncolytic herpes simplex virus-1 (HSV)) + radiation	NCT02457845

Status number shows: 1 = Active, not recruiting; 2 = Active, not recruiting / has results; 3 = Enrolling by invitation; 4 = Recruiting; N/A = Not available.

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
