# Peer review of "Therapeutic Strategies for Overcoming Immunotherapy Resistance Mediated by Immunosuppressive Factors of the Glioblastoma Microenvironment"

_cancers, 2020, doi:10.3390/cancers12071960_

Round 1

Reviewer 1 Report

Authors have presented -

(1) a comprehensive review and discussion of various immunosuppressive factors that are at play in GBM microenvironment.

(2) how targeting some of these mechanisms could potentially pave the way for successful reversal of the immunosuppressive nature of the tumour. 

(3) With regards to correlation of immunosuppression to genomic profile of the resected tumours, authors have briefly included PTEN mutation status along with M2 macrophage infiltration to have an affect on therapeutic effect of immune checkpoint inhibitors (ICIs)  (lines 164-165).

Similarly can the authors include more molecular indicators that might affect the immunosuppressive behaviour of GBMs either positively or negatively. They  might include research from other types of tumour as well.  Some recent papers are included below,

Liu X, Wang Y, Lu H, et al. Genome-wide analysis identifies NR4A1 as a key mediator of T cell dysfunction. Nature. 2019;567(7749):525-529.

Wang Y, Shen Y, Wang S, Shen Q, Zhou X. The role of STAT3 in leading the crosstalk between human cancers and the immune system. Cancer Lett. 2018;415:117-128. 

Can authors pay any specific thought on WHO defined molecular subtypes of GBM and their extend of immunosuppression, and which specific immunosuppressive factors might be at play in specific molecular subtypes. Any thought or discussion on this aspect?

Minor corrections-

(1) Lines 131-137 - Rewrite them more clearly. (more clear expression required)  

(2) Lines 144-148 - Rewrite them more clearly. (Long sentences and grammar needs correcting) 

(3) Line 173 Regulatory T cells (Tregs) and not "Regulatory Tregs"

Best Wishes,

Author Response

Reviewer 1

Suggestion1. Similarly can the authors include more molecular indicators that might affect the immunosuppressive behavior of GBMs either positively or negatively. They might include research from other types of tumour as well. Some recent papers are included below, Liu X, Wang Y, Lu H, et al. Genome-wide analysis identifies NR4A1 as a key mediator of T cell dysfunction. Nature. 2019;567(7749):525-529. Wang Y, Shen Y, Wang S, Shen Q, Zhou X. The role of STAT3 in leading the crosstalk between human cancers and the immune system. Cancer Lett. 2018;415:117-128.

-> Answer1. We thank the reviewer for valuable comments and helpful suggestions. We have taken all these comments and suggestions into account and they have improved our manuscript considerably. We also thank the reviewer for sharing articles with valuable information, which has been added in the text (lines 84 and 113), and corresponding references have been added to the manuscript.

Suggestion2. Can authors pay any specific thought on WHO defined molecular subtypes of GBM and their extend of immunosuppression, and which specific immunosuppressive factors might be at play in specific molecular subtypes. Any thought or discussion on this aspect?

-> Thank you for your thoughtful suggestion. The difference of the immunosuppression status between IDH wildtype- and IDH mutant type GBM may be an important issue. 2-HG is highly produced and accumulated in the tumor microenvironment in IDH mutant type GBM. 2-HG directly or indirectly suppresses the effector function of immune cells and their infiltration into the tumor microenvironment. The research presented at ASCO2020 showed that microglia is predominantly responsible for immunosuppression in the environment of IDH mutant type GBM, while bone marrow-derived macrophages may be dominant in IDH wildtype GBM (DOI: 10.1200/JCO.2020.38.15_suppl.2509, Journal of Clinical Oncology 38, no. 15_suppl (May 20, 2020) 2509-2509.). We speculate that chemokine release regulation by environmental 2-HG affects these immune cell population, and these differences will be important in the development of treatment strategies using immunotherapy for GBM patients. However, we cannot add these speculations in the text since they are not proved scientifically.

Minor corrections-

(1) Lines 131-137 - Rewrite them more clearly. (more clear expression required) 

-> Thank you for your suggestion. These sentences have been rewritten.

(2) Lines 144-148 - Rewrite them more clearly. (Long sentences and grammar needs correcting)

-> Thank you for your suggestion. This long sentence has been divided into two sentences.

(3) Line 173 Regulatory T cells (Tregs) and not "Regulatory Tregs"

-> It has been corrected.

We are really grateful for your peer review and thoughtful comments.

Reviewer 2 Report

The authors present an extensive review on GBM, focusing on immunomodulation and recurrent GBM. The rapidly growing knowledge in this field definitely warrants a review-article. The authors summarize immunomodulatory pathways and possible targets as well as ongoing trials. The review is well written and backed with citations from literature, so I have no comments and recommend publication in Caners.

Author Response

The authors present an extensive review on GBM, focusing on immunomodulation and recurrent GBM. The rapidly growing knowledge in this field definitely warrants a review-article. The authors summarize immunomodulatory pathways and possible targets as well as ongoing trials. The review is well written and backed with citations from literature, so I have no comments and recommend publication in Caners.

-> We thank the reviewer for the reviewing our manuscript and the positive comments.

Reviewer 3 Report

The review article "Therapeutic strategies for overcoming immunotherapy resistance mediated by immunosuppressive factors of the tumor microenvironment in glioblastoma" by Miyazaki et al, describes the interactions of glioblastoma with its microenvironent and the efforts to therapeutically exploit these interactions.

The article is well-written and informative.

I would suggest a minor revision, which involves presentation of the results in table 1. Column lengths could be changed as follows:

A) the last column (NCT number) could appear in one line and

B) the columns "status" and "phase" could appear as one column (Status/phase). The different status could also be defined by numbers, which would be explained at the end of the table (For example 1/I, where 1=Active, Not recruiting). This intervention will avoid repetition of phrases and give space to expand the column "Study title".

The data from the table should be further discussed, by expanding unit "7.ongoing clinical trials" with a new paragraph, including a comprehensive summary of the table 1 contents.  

Author Response

The review article "Therapeutic strategies for overcoming immunotherapy resistance mediated by immunosuppressive factors of the tumor microenvironment in glioblastoma" by Miyazaki et al, describes the interactions of glioblastoma with its microenvironent and the efforts to therapeutically exploit these interactions.

The article is well-written and informative.

-> We thank the reviewer for valuable comments and helpful suggestions. We have taken all these comments and suggestions into account and they have improved our manuscript considerably.

I would suggest a minor revision, which involves presentation of the results in table 1. Column lengths could be changed as follows:

Suggestion 1. The last column (NCT number) could appear in one line and the columns "status" and "phase" could appear as one column (Status/phase). The different status could also be defined by numbers, which would be explained at the end of the table (For example 1/I, where 1=Active, Not recruiting). This intervention will avoid repetition of phrases and give space to expand the column "Study title".

-> Thank you for your kind suggestion on how to correct the notation in the table. It has been corrected according to your suggestions.

Suggestion 2. The data from the table should be further discussed, by expanding unit "7.ongoing clinical trials" with a new paragraph, including a comprehensive summary of the table 1 contents.

-> Thank you for your fantastic suggestion. Unit "7.ongoing clinical trials" has been expanded according to your suggestion.

We are really grateful for your peer review and thoughtful comments.